# Language Outcomes of Children Born Very Preterm in Relation to Early Maternal Depression and Anxiety

**DOI:** 10.3390/brainsci13101355

**Published:** 2023-09-22

**Authors:** Sisan Cuervo, Nancy Creaghead, Jennifer Vannest, Lisa Hunter, Chiara Ionio, Mekibib Altaye, Nehal A. Parikh

**Affiliations:** 1Department of Communication Sciences and Disorders, University of Cincinnati, Cincinnati, OH 45267, USA; sisan.cuervo@msj.edu (S.C.); creaghea@ucmail.uc.edu (N.C.); vannesjr@ucmail.uc.edu (J.V.); 2Communication Sciences Research Center, Cincinnati Children’s Hospital Medical Center, Cincinnati, OH 45229, USA; 3Center for Prevention of Neurodevelopmental Disorder, Cincinnati Children’s Hospital Medical Center, Cincinnati, OH 45229, USA; mekibib.altaye@cchmc.org (M.A.); nehal.parikh@cchmc.org (N.A.P.); 4Department of Psychology, Catholic University of the Sacred Heart, 00168 Milano, Italy; chiara.ionio@unicatt.it; 5Division of Biostatistics and Epidemiology, Cincinnati Children’s Hospital Medical Center, Cincinnati, OH 45229, USA; 6Perinatal Institute, Cincinnati Children’s Hospital Medical Center, Cincinnati, OH 45229, USA; 7Department of Pediatrics, University of Cincinnati College of Medicine, Cincinnati, OH 45229, USA

**Keywords:** very preterm, outcomes, language, maternal depression, anxiety

## Abstract

Unaddressed maternal psychological distress within the first year postpartum is known to have numerous negative consequences on the child’s developmental outcomes, including language acquisition. This study examined the relationship between early maternal psychosocial factors and the language outcomes of children born very preterm (VPT; ≤32 weeks gestational age). It used data from the Cincinnati Infant Neurodevelopment Early Prediction Study, an ongoing National-Institutes-of-Health-funded prospective, multicenter cohort investigation of VPT infants. A total of 243 (125 boys; 118 girls) children born VPT (M = 29.03 weeks of gestation; SD = 2.47) and their corresponding 207 mothers (34 with multiple infants) were included in this study. We did not find an association between maternal depression or anxiety and Bayley-III (M = 92.3, SD = 18.9) language scores. Additionally, maternal grit and self-efficacy did not modify the relationship between depression and anxiety and language scores. A higher level of maternal education and infant female sex were significantly associated with higher language scores. While preterm birth typically results in higher rates of depression and anxiety for parents, the findings suggest that maternal depression, anxiety, and grit and the self-efficacy of the mothers in this sample did not relate to the language development of their children, independent of maternal education and infant female sex.

## 1. Introduction

The mental health and resilience of mothers during the postpartum period are important for their children’s overall development, including language development [1]. Early language experiences via mother–infant interactions are foundational for children’s social and cognitive growth, as well as their future academic success. Unaddressed maternal psychological distress within the first year postpartum is known to have numerous negative consequences on the child’s developmental outcomes, including language acquisition [1,2,3,4,5]. A very preterm (VPT; ≤32 weeks gestational age) birth creates high levels of psychological distress for mothers, and approximately 40% of VPT infants develop speech and language disorders [6,7,8,9,10,11].

Children born preterm are at a high risk for delays in cognitive and motor skills. Additionally, delays have been reported in the areas of receptive and expressive language across measures of vocabulary size, length of utterance, complexity of language, and phonological short-term memory [7,12,13,14]. Reidy et al. (2013) reported significantly poorer performances of children born preterm on measures of phonological awareness, semantics, grammar, discourse, and pragmatics when compared to controls born at term [15]. Woods et al. (2014) described the language trajectory of children between 1 and 5 years old who were born VPT, with the aim of defining predictive values of early language testing and the stability of language development [16]. Their results indicated a high incidence of specific language impairments amongst their cohort. These language difficulties can impact functional communication and future academic success.

Maternal postpartum psychological distress, defined as presenting mild to severe levels of depression and/or anxiety, can further negatively impact a child’s language outcomes [5,17,18,19]. A meta-analysis by Rogers et al. (2020) on the long-term developmental outcomes of children whose mothers reported experiencing depression and anxiety (combined) suggests that the mother’s experience is adversely associated with the child’s outcomes in the areas of social–emotional, cognitive, language, motor, and adaptability development [20].

In a longitudinal study, Quevedo et al. (2012) reported that the children of mothers who were postnatally depressed when their infant was between one and three months of age had delayed language acquisition at 12 months of age [21]. Brennan et al. (2000) reported that the severity and chronicity of maternal depressive symptomatology were related to lower vocabulary scores in children at 5 years of age [22]. In another longitudinal study, Aoyagi et al. (2019) concluded that exposure to late-onset (5–12 weeks) maternal postpartum depression is associated with a delay in neurodevelopment, reflected by a persistent decline in the rate of expressive language outcomes at four different time points (18, 24, 32, and 40 months) through infancy and early childhood [23].

Experiencing a preterm birth suddenly interrupts the mental representations and expectations that mothers construct during pregnancy, potentially turning the birth into a traumatic event that generates increased levels of psychological distress [24]. The prevalence rates of postpartum distress in mothers who experience a preterm birth range from 23 to 40%, which represents a significant increase from the range of 8–19% among new mothers of infants born at term [18,19,25,26,27]. Gjerdingen et al. (2011) reported that the proportion of mothers with high levels of depressive symptomatology was the greatest between birth and one month postpartum, indicating a possible increase in maternal distress during this time [28]. As with depression, high levels of anxiety have been reported in approximately 25% of mothers following preterm labor, compromising the mother’s functioning and interactions with the infant [18,19,29]. Additionally, there are environmental factors that may contribute to high levels of maternal distress such as living in poverty, not having social support (i.e., family, spouse, friends), being unemployed, and experiencing a child’s medical uncertainties [30].

Coping with increased levels of postpartum distress varies based on maternal environmental factors (i.e., social support, prior mental health problems, care provided in the NICU), as well as internal factors such as character traits. In adults, character traits such as grit and self-efficacy act as distress buffers and have been used to predict coping mechanisms when dealing with daily hassles, as well as adaptation skills after experiencing stressful life events [31,32]. Bandura (1982) defines self-efficacy as an individual’s judgments about how their parenting abilities affect their cognitive and emotional reactions to stress which, in turn, impact early parent–child interactions. Studies have linked high levels of self-efficacy to increased parental competence and reduced levels of psychological distress [32]. Studies of parents of preterm infants have demonstrated a positive relationship between self-efficacy beliefs and parenting behaviors [33]. A similar character trait that refers to an individual’s perseverance and motivation toward a specific task or goal is grit [31]. Studies on grit report a positive relationship between grit, long-term psychological well-being, and life satisfaction in adults [34]; however, it is a trait that is under-explored in mothers following a preterm birth.

Given the high levels of distress reported by mothers of VPT infants and the impact that they may have on the children’s language outcomes, this study aims to (1) analyze the relationship between maternal depression and children’s language outcomes, (2) analyze the relationship between maternal anxiety and children’s language outcomes, and (3) understand how the self-efficacy and grit scores of mothers influence the relationship between maternal distress and language development in VPT children at 24 months corrected age.

## 2. Materials and Methods

Family recruitment was conducted and informed written consent was collected by neonatal intensive care unit nurses in five hospitals of a midwestern state of the United States between September 2016 and November 2019. This occurred between the birth of an infant and prior to its discharge from hospital. A total of 243 children born VPT (M = 29.02 weeks of gestation; SD = 2.48) and their corresponding 207 mothers (34 with multiple infants) who were part of the CINEPS cohort were included in this study. The children’s gestational ages ranged from <28 weeks’ gestation (n = 89) and 29 to 32 weeks’ gestation (n = 154). Infants with cyanotic heart disease, known chromosomal or congenital anomalies affecting the central nervous system, or infants hospitalized and mechanically ventilated past the 44-week period were ineligible for participation in the study. See Table 1 for demographic information.

### 2.1. Procedures

The study’s first timepoint visit for the mothers and children occurred approximately at the infant’s 40 weeks postmenstrual age. At this visit, the mothers provided demographic information and filled out the Patient Health Questionnaire (PHQ-9) to measure levels of depression, the Patient Reported Outcomes Management Information System (PROMIS) for Anxiety, the GRIT Scale, and the Generalized Self-Efficacy Scale (GSES). Given the language growth seen around 24 months of age, all 243 children were assessed using the Bayley Scales of Infant and Toddler Development, Third Edition (Bayley-III), for a corrected age of 24 months by examiners who were blinded to the mothers’ depression and anxiety status. The Bayley-III provides norm-referenced index scores for cognition, language, and motor indices, as well as subscale scores for expressive language, receptive language, and fine motor and gross motor abilities.

### 2.2. Data Analyses

Two generalized linear mixed models (LMMs) were constructed to investigate the association between (1) the children’s Bayley-III language scores at 24 months corrected age and the maternal PHQ-9 scores at approximately 40 weeks postmenstrual and (2) the children’s Bayley-III language scores at 24 months corrected age and the maternal PROMIS Anxiety scores at approximately 40 weeks postmenstrual. The PHQ-9 and PROMIS Anxiety scores were analyzed as continuous variables and dichotomized by severity to describe any possible relationship attributed to skewed data. The relationships between maternal distress and childhood language outcomes were studied after adjusting for potential confounders identified by the literature as either being associated with language skills in early childhood or as impacting language outcomes following exposure to maternal distress. Preliminary analyses considered family income, race, maternal education, marital status, the number of children living in the household, and the child’s sex and gestational age. Confounders that did not have a significant relationship with language were removed from the models. The final models included two behavioral confounders (maternal education and the sex of the child) previously shown to influence language development in VPT infants [12,23,35].

To analyze the moderating effect of maternal self-efficacy and grit on the relationship between maternal distress and language development, the following interaction terms were included in the initial models (Grit Scale x PHQ-9; GSES x PHQ-9; Grit Scale x PROMIS Anxiety scores; GSES x PROMIS Anxiety scores). None of the interaction terms were found to moderate the relationship between maternal distress and language development. However, given the importance of these two variables in relationship to the child’s language outcomes, they were included in the two LMMs as covariates. The cohort includes multiple births (37 twins and 2 triplets); thus, a random intercept was included to account for the potential correlation of responses. The criterion for significance for all analyses was set at *p* < 0.05.

## 3. Results

At the 40-week postmenstrual period, the mothers’ PHQ-9 scores (M = 4.01, SD = 4.14) indicated that 66.6% (n = 138) of the sample had minimal depressive symptoms, 23.6% (n = 49) had mild depression, 5.7% (n = 12) had moderate depression, and 4.1% (n = 8) had severe depression. The PROMIS Anxiety scores (M = 50.4, SD = 10.1) indicated that 62.3% (n = 129) of the mothers were not experiencing anxiety or emotional distress, 18.4% (n = 38) were experiencing mild anxiety, 16.4% (n = 34) were experiencing moderate anxiety, and 2.9% (n = 6) were experiencing severe anxiety. At the same time point, the maternal Grit Scale scores were M = 3.82 (the highest score is 5) and SD = 0.53, indicating below-average levels of maternal grit. The GSES scores were M = 33.81 (the highest score is 40) and SD = 4.39, indicating that the mothers had average levels of a generalized sense of self-efficacy. At 24 months corrected age, the children’s Bayley-III language scores (M = 92.3, SD = 18.9) indicated that 69.5% (n = 169) of the children in the sample presented language skills within normal limits, 18.5% (n = 45) had mild delays (<85 standard score), and 11.9% (n = 29) had moderate-to-severe delays (<70 standard score).

### 3.1. Maternal Depression and Child’s Language

The initial LMM of the relationship between maternal depression and language outcomes demonstrated no significant association between the mothers’ PHQ-9 scores and the children’s Bayley-III language outcomes (*p* = 0.85). The results reported are based on continuous data since similar results were observed when the PHQ-9 scores were dichotomized into severity groups. The levels of grit (*p* = 0.07) and self-efficacy (*p* = 0.24) in the mothers did not modify the association between levels of maternal depression at 40 weeks postmenstrual and the children’s language scores at 24 months corrected age. Both the child’s female sex (t = 4.41, df = 231.04, *p* = <.001) and the mother’s level of education (t = 7.74 df = 213.43, *p* = <.001) were significantly associated with the language outcomes, as presented on Table 2.

### 3.2. Maternal Anxiety and Child’s Language

A second LMM examining the relationship between maternal anxiety and children’s language outcomes demonstrated no significant association between the mothers’ PROMIS Anxiety scores and the children’s Bayley-III language outcomes (*p* = 0.74). The results reported are based on continuous data since similar results were observed when the PROMIS Anxiety scores were dichotomized into severity groups. The levels of grit (*p* = 0.07) and self-efficacy (*p* = 0.26) in the mothers did not modify the association between levels of maternal depression at 40 weeks postmenstrual and the children’s language scores at 24 months corrected age. As in the first model, both the child’s female sex (t = 4.41, df = 231.27, *p* = < 0.001) and the mother’s level of education (t = 7.73, df = 213.32, *p* = < 0.001) were significantly associated with the language outcomes, as presented in Table 3.

## 4. Discussion

This study examined the relationships between the maternal depression and anxiety of mothers who experienced VPT births and the language development of their children at24 months corrected age. Furthermore, this study investigated whether the mothers’ self-efficacy and grit influenced that relationship. Contrary to our hypotheses, self-reported maternal psychological distress was not associated with the children’s language outcomes. Even though high levels of maternal depression and anxiety have been demonstrated to adversely impact the language outcomes of children born VPT [21,23], our findings may be explained by methodological factors such as (a) the instrument used to measure language outcomes, (b) family-centered care in the NICU, (c) the timing of the measurement of maternal distress relative to the measurement of the child’s language outcomes, and (d) maternal contextual factors and the range of depression/anxiety scores within our sample of mothers. Our study presents unique strengths that can inform future studies and clinical practices.

### 4.1. Language Assessment Tool

Previous studies used a variety of language assessment tools, including the Bayley-III, to demonstrate that exposure to postpartum distress increases the risk of overall language delays at 12 months and is associated with poorer expressive language skills in children aged 18 months [21,23]. However, as discussed by Aoyagi and Tsuchiya (2019), different language measures may yield conflicting results when differentiating receptive, expressive, and/or overall language skills [1]. The Bayley-III is a commonly used standardized tool to assess early development. However, the language index of the Bayley-III may underestimate the rates of language delays and impairments in children born preterm [16,36,37].

Given the complexity of language development during the first two years of life and its relationship to environmental factors, we recommend using the Communication and Symbolic Behavior Scales (CSBS) and the MacArthur-Bates Communicative Development Inventories (CDI) to provide more comprehensive measures of language outcomes. The CSBS measures early communication skills as well as gestures, positive affect, gaze shifts, communicative functions, and the rate of communication [38]. The CDI is a parent report instrument that measures early language abilities such as receptive and expressive vocabulary, gestures, and grammar [39]. These tools have empirical research demonstrating the sensitivity and specificity of measuring language outcomes at 24 months corrected age, and are both being utilized in the ongoing follow-up of a subset of this cohort. Additionally, the CDI could provide parents with specific language-stimulation models to be incorporated in the home environment regardless of maternal characteristics. For example, the CDI utilized communication temptations to elicit communication from young children, a strategy that can be carried over into the home environment by the mothers.

### 4.2. Family-Centered Care Supports in the NICU

Given the importance of family-centered care for VPT infants, the minimal-to-mild levels of depression and anxiety reported by the mothers in our cohort may be moderated by the levels of support they received in the NICU before discharge and/or during follow-up visits. Based on family-centered care literature and the results from 103 neonatal nurses surveyed, Weber et al. (2021) discussed the need for and benefits to increasing peer-to-peer support groups for families in NICUs, mental health services, universal distress screeners, and further education for staff members who are part of the team [40]. The current study recruited families from five NICUs in a midwestern state of the United States. All five facilities aim to implement family-centered care for infants in the NICU and seek to provide parent-to-parent support for all families. Additionally, each family receives follow-up care from the NICU clinic based on their child’s individual needs. Psychologists and social workers are available as needed on a case-by-case basis both in the NICU and after discharge. As part of their enrollment in the CINEP Study, all families received follow-up visits, including a consultation with a neonatologist at a postmenstrual age of 41 weeks when they came in for their first study visit, followed by visits at 3, 12, 24, and 36 months the corrected ages.

### 4.3. Timing

Variations in the severity and chronicity of maternal depression and anxiety have also been shown to play a role in children’s developmental outcomes. Mothers with minimal-to-mild levels of depression, as well as those with shorter episodes of distress, have been shown to provide more supportive language-learning environments characterized by an increased quantity and/or quality of maternal vocal input than mothers with more severe symptoms and those whose symptoms are more chronic [41]. In our sample, maternal distress was measured via the PHQ-9 and the PROMISE Anxiety at 40 weeks postmenstrual age. The time between birth and the first study timepoint varied per family depending on the gestational age of the infant and the time spent in the NICU before discharge.

The proportion of mothers with elevated levels of depression was the highest between zero and four weeks postpartum [28]. In our study, the time frame between birth and when the degree of maternal distress was measured ranged between 5 and 17 weeks (M = 11 weeks postpartum); therefore, the period in which the mothers demonstrated acute distress symptomatology may have passed. However, to our knowledge, there is no evidence to suggest that the level of distress experienced by a mother only during the first four weeks postpartum impacts the child’s language trajectory. Measuring maternal distress levels at different time points (i.e., at 3, 6, 10, 14, and 24 months), as previously carried out by Prenoveau et al. (2017), may be beneficial when further exploring the relationship between distress and child language outcomes [42]. A positive outcome of our study is that we did not find that chronic levels of depression and/or anxiety were related to the children’s language outcomes in our sample.

### 4.4. Contextual Factors and the Range of Psychological Distress

Levels of maternal psychological distress in our sample were relatively low, with 90.3% (n = 187) of the mothers self-reporting minimal-to-mild depressive symptoms and 80.6% (n = 167) self-reporting mild to no anxiety or emotional distress. Mothers in minority groups and those living in poverty who have experienced higher numbers of adverse experiences are three times more likely to develop postpartum distress than those not exposed to adverse events [42]. Personal resiliency factors such as grit and self-efficacy are modifiable in mothers and can serve as buffers for those experiencing high levels of depression and anxiety. When working with VPT children and their families, modifiable individual characteristics should be considered key factors influencing early mother–child language interactions.

In this study’s sample, 82.4% of the mothers were more socioeconomically advantaged, 75% had some college experience or had completed a degree, and 68.1% reported as Caucasian. These demographic characteristics indicate that our sample might not exhibit the range of contextual factors that are known to highly interact with levels of maternal stress and anxiety (i.e., low SES, reduced maternal education, culturally and linguistically diverse minorities). An evident strength of the parent study is the relatively low (12%) rate of attrition, which allowed us to gather longitudinal data from many of the families.

The present study accounted for the twin/triplet birth factor, which is an important variable when studying the VPT population. Our findings cannot be generalized to other caregivers, including the fathers of VPT infants, since they may experience different stressors influencing levels of depression and/or anxiety. Given that 90.3% of the mothers in our sample reported minimal-to-mild levels of depression and 80.6% reported mild-to-zero levels of anxiety, future studies should include a larger group of mothers experiencing moderate-to-severe psychological distress and consider a further analysis of the relationship between maternal anxiety, depression, and children’s language outcomes. The use of self-report scales of depression/anxiety rather than clinical diagnoses may have been a limitation; however, we acknowledge the importance of screening NICU families and providing adequate mental health services or referrals when needed. Future studies should consider structured clinical interviews and/or a more in-depth standardized assessment by a mental health professional to determine the status of distress.

## 5. Conclusions

In conclusion, this study suggests that maternal depression, anxiety, and grit and the self-efficacy of the mothers in this sample did not relate to their VPT children’s language development, independent of maternal education and infant female sex. Further studies like this one that use interprofessional collaborations among neonatologists and speech–language pathologists (SLPs) will provide essential roadmaps to better serve this population.

## Figures and Tables

**Table 1 brainsci-13-01355-t001:** Sociodemographic and participant characteristics.

Children	N = 243
Gestational age—weeks (M, SD)	29.03 (2.47)
Sex	
Male	51.4% (n = 125)
Female	48.6% (n = 118)
Mothers	N = 207
Multiple gestations	
Singletons	80.6% (n = 167)
Twins	18.3% (n = 38 *)
Triplets	1.1% (n = 2)
Race	
White	68.1% (n = 141)
Black or African American	23.6% (n = 49)
Asian	2.8% (n = 6)
Biracial	2.4% (n = 5)
American Indian or Alaskan Native	<1% (n = 1)
Other	1.4% (n = 3)
No Response	<1% (n = 2)
Education	
Less than high school diploma or GED	6.2% (n = 13)
High school diploma or GED	18.8% (n = 39)
Some college	29.4% (n = 61)
College degree	22.7% (n = 47)
Some graduate school	3.6% (n = 7)
Graduate degree or higher	19.3% (n = 40)
Relationship Status	
Married	53.6% (n = 111)
In a relationship, living with partner	20.7% (n = 43)
In a relationship, not living with partner	9.1% (n = 19)
Separated/Divorced	3.6% (n = 7)
Single	13% (n = 27)
Other Children in Household	
No	5.4% (n = 11)
Yes	94.6% (n = 196)

* 6 mothers who reported multiple gestations only had one child enrolled in the study.

**Table 2 brainsci-13-01355-t002:** Results of a linear mixed model of children’s language outcomes and PHQ-9 Depression Scale scores.

Parameter	Bayley-III Language			
	Estimate	SE	Test (df)	*p*
Intercept	62.53	11.45	t = 5.46 (203.76)	<0.001
PHQ-9	−0.05	0.28	t = −0.18 (220.58)	0.85
Grit Scale	−4.00	2.26	t = −1.77 (207.55)	0.07
GSES	0.31	0.27	t = 1.15 (205.91)	0.24
Sex (female)	9.17	2.07	t = 4.41 (231.04)	<0.001 *
Highest Level of Maternal Education	5.88	0.75	t = 7.74 (213.43)	<0.001 *

* Significant at *p* < 0.05; PHQ-9: Patient Health Questionnaire (depression); GSES: Generalized Self-Efficacy Scale.

**Table 3 brainsci-13-01355-t003:** Results of a linear mixed model of children’s language outcomes and PROMIS Anxiety Scale scores.

Parameter	Bayley-III Language			
	Estimate	SE	Test (df)	*p*
Intercept	64.52	13.70	t = 4.71 (210.71)	<0.001
PROMIS Anxiety	−0.03	0.11	t = −0.32 (212.34)	0.74
Grit Scale	−4.01	2.23	t = −1.79 (207.44)	0.07
GSES	0.30	0.27	t = 1.10 (208.31)	0.26
Sex (female)	9.15	2.07	t = 4.41 (231.27)	<0.001 *
Highest Level of Maternal Education	5.91	0.76	t = 7.73 (213.32)	<0.001 *

* Significant at *p* < 0.05; PROMIS Anxiety: Patient Reported Outcomes Management Information System. Anxiety Short Form; GSES: Generalized Self-Efficacy Scale.

## Data Availability

The data that support the findings of this study are available upon request from the author, Lisa Hunter. The data are not publicly available due to privacy and ethical restrictions.

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
