# Peer review of "Language Outcomes of Children Born Very Preterm in Relation to Early Maternal Depression and Anxiety"

_brainsci, 2023, doi:10.3390/brainsci13101355_

Round 1

Reviewer 1 Report

Authors describing an important study examining factors associated with language delay in preterm infants. The study holds promise but Authors are cautioned to soften recommendations that are based on their study's findings (or lack there of). Questions/comments below are suggested to help improve the manuscript.

Introduction

Major: Although very well written and succinct, the introduction provides very little theory or rationalization linking mothers' psychological distress and their preterm infant's language delay. Additionally, Authors seem to group depression, anxiety, etc. together implying that they all play the same role in speech/language delays. The introduction also provides sparce discussion on the potential consequences these language delays may have for these children. Lastly, but importantly, the study's hypotheses are completely exploratory. No case is made or argument provided (based on prior literature) to help inform predictions. 

Minor: Why 24mos? I assume because language is your outcome but a quick explanation is needed.

Methods & Results

Major: Describe specific language outcomes assessed by the Baileys.

Did any interaction approach significance? Some researcher argue that interactions that are p<.1 can be probed.

Minor: Explain why some of the confounds dropped and why LMM and not regression.

Were all interaction terms put in the same model? Was a power analyses performed given the number of variables?

Discussion

Major: Although recommendations may be valid, they don't stem from this study's results as you did not compare language measures nor had a control group that did not experience a family centered NICU. Language here must be soften or reframed.

Where are the models overall fit indexes?

Why didn't Authors discuss reasons for why education and sex were significant?

Minor: Self-efficacy approached significance.

Reviewer 2 Report

The paper presents an empirical study on language outcomes of children born very preterm in relation to early maternal depression and anxiety.

The manuscript is well written and understandable.

The research topic is of interest for both research and practice.

Important strengths include the use of a sophisticated database and the detailed investigations.

The paper could nicely fit into the Special Issue Neurodevelopmental Disorders and Early Language Acquisition.

The absence of significance for the main hypotheses could be elaborated in more detail. Is it due to methodological reasons or statistical power?

The authors could more broadly discuss the pattern of results in light of other performance domains of the targeted population of very early born children.

Did the interactions with the father had a role (perhaps moderation effect)?

The practical relevance could be discussed with more detailed examples from everyday life.

Round 2

Reviewer 1 Report

Author's addressed previous comments and improved their manuscript.